# Learning from and Leveraging Multi-Level Changes in Responses to the COVID 19 Pandemic to Facilitate Breast Cancer Prevention Efforts

**DOI:** 10.3390/ijerph18136999

**Published:** 2021-06-30

**Authors:** Deborah J. Bowen, Kelly E. Rentscher, Amy Wu, Gwen Darien, Helen Ghirmai Haile, Jeanne Mandelblatt, Marion Kavanaugh-Lynch

**Affiliations:** 1Department of Bioethics and Humanties, University of Washington School of Medicine, 1959 NE Pacific St., P.O. Box 357120, Seattle, WA 98195, USA; hghaile@uw.edu; 2Cousins Center for Psychoneuroimmunology, Semel Institute for Neuroscience and Human Behavior, University of California, 300 Medical Plaza, Suite 3156, Los Angeles, CA 90095, USA; krentscher@ucla.edu; 3California Breast Cancer Research Program, University of California Office of the President, 1111 Franklin Street Oakland, Oakland, CA 94607, USAmarion.kavanaugh-lynch@ucop.edu (M.K.-L.); 4National Patient Advocate Foundation, 1100 H Street, NW, Suite 710, Washington, DC 20005, USA; gwen.darien@npaf.org; 5Department of Oncology, Cancer Prevention and Control Program Georgetown-Lombardi Comprehensive Cancer Center and Department of Medicine, Georgetown University Medical Center, 3300 Whitehaven Street NW, Washington, DC 20007, USA

**Keywords:** COVID-19, breast cancer, review paper, cancer prevention

## Abstract

The coronavirus pandemic (COVID-19) has had multilevel effects on non-COVID-19 health and health care, including deferral of routine cancer prevention and screening and delays in surgical and other procedures. Health and health care use has also been affected by pandemic-related loss of employer-based health insurance, food and housing disruptions, and heightened stress, sleep disruptions and social isolation. These disruptions are projected to contribute to excess non-COVID-19 deaths over the coming decades. At the same time municipalities, health systems and individuals are making changes in response to the pandemic, including modifications in the environmental to promote health, implementation of telehealth platforms, and shifts towards greater self-care and using remote platforms to maintain social connections. We used a multi-level biopsychosocial model to examine the available literature on the relationship between COVID-19-related changes and breast cancer prevention to identify current gaps in knowledge and identify potential opportunities for future research. We found that COVID-19 has impacted several aspects of social and economic life, through a variety of mechanisms, including unemployment, changes in health care delivery, changes in eating and activity, and changes in mental health. Some of these changes should be reduced, while others should be explored and enhanced.

## 1. Introduction

The COVID-19 pandemic has had a complex and multi-faceted impact on health [1,2]. Non-COVID-19-related health care has been disrupted, including deferral of routine cancer prevention and screening and delays in surgical and other procedures. Health and health care use has also been affected by pandemic-related loss of employer-based health insurance, food and housing disruptions, and heightened stress, sleep disruptions and social isolation. These disruptions are projected to contribute to excess non-COVID-19 deaths, including cancer deaths over the coming decades [3,4,5,6]. At the same time municipalities, health systems and individuals are making changes in response to the pandemic, including modifications in the environmental to promote health, implementation of telehealth platforms, and shifts towards greater self-care and using remote platforms to maintain social connections. All of these changes have the possibility of being leveraged to improve health and decreasing the burden of cancer.

In this paper, we use a multi-level biopsychosocial model to summarize the potential impact of the COVID-19 pandemic on breast cancer incidence and discuss the ways to leverage pandemic-related changes to reduce the risk of breast cancer. Finally, we propose a research agenda to document the effects of the COVID-19-related changes on breast cancer risk pathways and guide the development of interventions to provide evidence to support maintenance and expansion of effective strategies.

### Conceptual Framework

Many factors have been identified as risk factors for breast cancer [7,8]. These risk factors occur and/or are influenced at multiple levels, including the built, social, natural, social, political and healthcare environment and individual and biological levels (Figure 1) [9]. We use this framework to guide our understanding of the effects COVID-19 pandemic-related changes in the environment and individual behavior on opportunities for and constraints on breast cancer prevention via effects on biological systems and/or interacting with underlying cellular and genetic risks for developing breast cancer. In this framework it is possible that effects of the infection on immune response could also have direct long-term effects on cancer rates, but this remains very speculative, and is beyond the scope of this paper. Given that these effects will likely develop over time, the careful study of time as it relates to the effects of COVID-19 is needed. We would like to note that this review does not cover the literature on COVID-19 and inflammation.

In the next sections we use this framework to highlight how changes in the environment and individual behavior might be leveraged to reduce risk of breast cancer and suggest areas for future research to confirm these effects and understand their biological underpinnings (Figure 1). As seen in this figure, several genetic and biological factors influence the overall process of breast cancer development, as well as temporal influences such as the timing of reproduction and aging. Given that older age has also been identified as a risk factor for COVID-19 infection and mortality (REF), future research can use this model to evaluate the role of COVID-19 and age in breast cancer risk.

## 2. Environmental-Level Pandemic-Related Changes and Breast Cancer Risk

The COVID19 pandemic has had influences on broad aspects of the environment and these, in turn, can each potentially increase or decrease breast cancer risk.

### 2.1. Changes in the Built Environment

The sudden closure of offices and businesses early in the pandemic sent millions of employed American’s home to conduct their work remotely. For this estimated 35% of the workforce [10], reports have described increases in physical activity facilitated by more discretionary time that people used to spend commuting [11]. “*People have been very resourceful in how to get more physical activity*,” says a weight loss physician. “*They track steps as they increase walking, use home exercise equipment, do calisthenics and access virtual workouts on the internet or via apps*” [12]. *USA Today* reported that sales of bikes and bike accessories nearly doubled from the previous years, as did trail use [13], while bikes, pools and camping gear sold out across the nation.

In addition, although there were closures of playgrounds and gyms, municipal changes such as the establishment of car-free streets for walking, biking, and skating encouraged more physical activity. A global survey on usage of public spaces during the pandemic reported that two-thirds of respondents are using public spaces, half of them daily, with a preponderance using neighborhood streets and sidewalks [14]. The same survey reported that two-thirds of respondents reported walking more since the shutdown. Additionally, 25% reported biking more. The authors state “*Whether through greater adoption of walking or biking, the pandemic may present opportunities to make healthy, sustainable ways of getting around a more integral part of our behaviors in the long-term*.” Studies have consistently shown that exercise of nearly any type and amount can reduce the risk of both pre- and post-menopausal breast cancer [15]. Maintenance of these new opportunities and venues for physical activity, as well as continued telecommuting to free time in the day, could be leveraged to reduce breast cancer risk.

Local governments and insurance companies have also partnered with mobile application developers to provide free access to mindfulness and meditation applications (e.g., Calm, Headspace, Ten Percent Happier, UCLA Mindful) and have offered virtual fitness classes [16]. Given that mindfulness meditation and other mindful movement practices (e.g., yoga, tai chi) have also been found to improve depressive symptoms, anxiety, and sleep quality, and reduce inflammation [17,18,19], increased access to mindfulness and meditation programs can decrease these recognized breast cancer risk factors.

### 2.2. Changes in the Natural Environment

One of the most notable early impacts of the COVID-19 pandemic were changes in the natural environment that arose from the sudden near cessation of petroleum-fueled transportation due to remote work, business closures and reduced air travel. These changes led to clearing of the air in cities and reduced air pollution. Average global air quality during lockdowns in 2020 improved relative to the same periods in 2019 [20]. Certain types of traffic-related air pollution, such as nitrogen dioxide (NO_2_) and nitrogen oxides (NO_x_) levels, both of which are proxies for traffic exposure, have been classified as carcinogens by the International Agency for Research on Cancer (IARC) and have been linked to breast cancer risk [21,22,23,24]. In addition, some components of traffic-related air pollution, including polycyclic aromatic hydrocarbons may act as endocrine disruptors [22,23,24]. These may have the most impact during the prenatal, pubertal and pregnancy windows of susceptibility. Thus, maintaining reduced traffic post-pandemic has the long-term potential to reduce population breast cancer rates.

### 2.3. Changes in the Social Environment

The coronavirus pandemic and resulting social distancing guidelines have led to many changes to the social environment, including disruptions to work and education routines, families moving in together, increases in informal or unpaid caregiving for children and ill family members, and social stressors such as tension and conflict with family members (e.g., crowded living situations, increased rates of domestic violence or other abuse), and decreased availability of social support and physical affection [25,26].

Some individuals have found greater social support and connection through the use of electronic platforms [27,28,29], which may help to offset some of the negative effects of the pandemic experience. For example, existing social networks focused on exercise have been re-organized for online formats, maintaining connections and making them more widely available. Other forms of entertainment have also been adapted to enhance social connections, ranging from online board- and card-game platforms, watch parties, book groups, and music and video software that allows individual performances to be merged into ensemble pieces. Ordinary congregational social events have continued on-line through virtual birthday parties, showers, weddings and holiday gatherings. For some families that have moved in together, the pandemic may increase connection and prevent or decrease feelings of loneliness in individuals who might otherwise be vulnerable, such as older and/or retired individuals. At the societal level, the shared experience of social isolation may also strengthen feelings of social cohesion, although it is unclear whether or how these feelings might change over time [30,31]. Thus, for some individuals, physical distancing during the pandemic may have facilitated reconnection, more frequent communication via video-conferencing, and increased feelings of closeness with friends and family. These changes can enhance well-being and decrease breast cancer risks through reduced inflammation (see Section 3.4 for a more detailed discussion).

### 2.4. Changes in the Economic and Geo-Political Environment

The U.S. unemployment rate has been up to 15%, the highest rate historically since the Great Depression [32]. As of February 2021 the U.S. had 10 million fewer jobs than it did pre-pandemic [33]. Half of the 10 million jobs were lost by women, the loss prompted by working mothers having to leave the workforce to take care of children and the household [34].

COVID-19 also created differences in economic impact by socioeconomic class. Individuals in upper middle class and upper-class households had options including telework, food delivery and could afford new ways of health and fitness such as home gyms. However, the pandemic led to job loss among groups already economically challenged [35]. Without the means to earn an income during lockdowns, many are unable to feed themselves and their families. This growing food insecurity is disproportionately affecting communities of color [36]. Difficulty paying for necessities such as rent or food and borrowing money or taking out a loan to make ends meet was a common experience. The rate of having suffered any one of these economic hardships was more than twice as high among Latino and Black adults than it was for white adults.

At the level of the geopolitical environment, the pandemic has highlighted long-standing structural inequities and racial stereotyping and discrimination [26,37]. These economic and geopolitical effects of the pandemic are likely to broadly create constraints on breast cancer risk reduction and prevention.

### 2.5. Changes in the Healthcare Environment

The spread of the virus and its health effects have necessitated dramatic changes to healthcare delivery. For example, in-person visits for non-COVID-19 care were immediately shut down in most care facilities to enable providers to reduce the spread of infection and increase capacity for COVID care. To compensate, the Centers for Medicare and Medicaid Services (CMS) lifted a variety of federal restrictions on telehealth services, and other insurers followed suit to cover telemedicine visits [38]. These changes to the medical environment have been lauded for having the potential to increase access to care while not compromising the quality of the patient experience [39,40,41]. It is possible or likely that insurance coverage for telemedicine may be reduced after the COVID pandemic has receded [42,43] or that added regulations will occur, but some form of expanded telehealth is likely to remain. A number of surveys show that Americans increasingly prefer telehealth to in person and plan to continue with telehealth post pandemic [44].

There are several ways in which telemedicine could improve breast cancer prevention and risk reduction activities. Primary care providers are the first line available via telehealth and they take the lead on changing health behaviors related to breast cancer prevention, such as smoking, obesity, and physical activity [45]. These behavioral changes are best achieved with brief, conversational contacts with patients to instruct, support and encourage and are well suited for telehealth. Telehealth can also expand access to mental health care [25,46], reducing stress and its links to breast cancer risk.

## 3. Individual-Level Pandemic-Related Changes and Breast Cancer Risk

In this section, we focus on individual level pandemic-related behaviors and perceptions that relate to known breast cancer risk factors: physical activity and sedentary behavior, diet, smoking and alcohol use, and sleep, psychological stress and perceived social isolation.

### 3.1. Physical Activity and Sedentary Behavior

The pandemic-related increases in physical activity described earlier in the paper may be driven in part by individual-level factors such as efforts to manage psychological stress and improve disrupted sleep patterns during the pandemic [47] Research suggests that some individuals have increased exercising at home and have opted for walking and biking rather than car travel [48]. For example, research conducted on a convenience sample of 604 Irish adults during COVID-19 restrictions found that activities including exercising, walking, gardening and pursuing hobbies were ranked as most enjoyable and were positively associated with positive affect [49]. However, for many, the pandemic has increased sedentary time that could increase risk for breast cancer [50].

### 3.2. Diet

Widespread restaurant closures and increased time at home provided the challenge and the opportunity for many people to purchase, prepare and consume foods at home. This has previously been shown to increase consumption of nutrient-dense foods as well as to decrease consumption of salt, sugar, fat and calories [51]. An internet survey comprised of a majority of U.S. respondents of normal- and over-weight adults revealed an overall increase in healthy eating due to less eating out and more cooking at home [52]. This increase was more pronounced among obese responders although one-quarter of the sample reported weight gain [52]. Two marketing surveys reported that almost a third of those surveyed planned to continue to cook at home once stay at home orders are lifted, especially among younger demographics [53,54,55]. In addition, an analysis of internet searches demonstrated an explosion of cooking-related searches [56] and a trend towards planting “pandemic gardens” [57]. Thus, there is the potential for this shift to have a positive impact on health [58,59,60]. and breast cancer risk through a reduction in obesity and increase in healthy eating [61].

Similarly, while cooking at home has been shown to translate to more healthful eating, it has not been uniformly linked to better eating habits during COVID-19. Anecdotally, many people are gaining “pandemic pounds.” This may be linked to stress, or to “snacking while bored.” With many businesses cutting back personnel, the remaining workforce has often had to take on more responsibilities on the job, which can also contribute to overeating. A cross-sectional survey in Poland [62] showed that half of the respondents, and especially those who were overweight or obese, were eating more or more frequently. Overweight or obese participants were more likely to gain weight whereas underweight participants tended to lose weight. Those who gained weight ate fewer fruits and vegetables and more meat, dairy, and fast foods. Similarly, in an Italian study, the perception of weight gain was observed in 48.6% of the population [63]. Since post-menopausal weight gain and obesity are associated with increase in breast cancer risk [64], pandemic-related weight gains could result in a future increase in breast cancer incidence.

### 3.3. Smoking and Alcohol Use

Research on smoking rates during the pandemic also suggest both increases and decreases. For instance, a recent study found that 35.6% of smokers reported an increased motivation to quit during the pandemic, and 22.9% reported an attempt to quit in order to reduce risk of negative outcomes from COVID-19 [65]. Data from smoking quit lines also showed an increase in phone and website registrations in March 2020 compared to the previous year [65]. Individuals’ increased motivation to quit smoking during the pandemic may therefore represent a “teachable moment” that could be leveraged in breast cancer prevention initiatives. However, several studies have found that some smokers increased (6.9–45%) their tobacco use during the pandemic [62,65,66,67], possibly due to boredom, stress, or depression associated with shelter-in-place orders, and/or economic and food insecurity [66,68]. Studies have also estimated that 13–27% of adults have increased their consumption of alcohol during the pandemic [25,62,67]. U.S. data indicated a spike in the general rate of drinking as a response to the COVID-19 pandemic [69,70] with alcohol sales skyrocketing 55% during the first few weeks of shelter-in-place [71]. Since smoking and alcohol increase risk of breast cancer [72], the pandemic may lead to future increases in breast cancer incidence.

### 3.4. Sleep, Psychological Stress and Perceived Social Isolation

The pandemic has also affected individual-level well-being, including sleep, stress and isolation. Sleep disruptions were reported among 38–41% of adults from March to April 2020 [67,73], including increases in the prevalence of clinical insomnia from prior to the pandemic [74,75]. Disruptions in daily routines and daylight exposure are also likely to disrupt circadian rhythms and further contribute to sleep disturbances [76,77]. When experienced over long periods, these types of circadian rhythm disturbances have been linked to increased breast cancer risk [78]. Circadian rhythm and sleep disruptions (e.g., night-shift work) have been identified as risk factors for breast cancer [79,80]. Likewise, sleep disturbances can have broad effects on inflammation and disease risk via activation of the stress-response system, although the direct link of sleep to breast cancer risk has not been established.

The COVID-19 pandemic has also been associated with increases in psychological stress related to depression and anxiety symptoms and perceived social isolation and loneliness [25,37,67,81]. For instance, the Centers for Disease Control (CDC) Household Pulse Survey conducted from April to June 2020 found that 31% of adult respondents had symptoms of a depressive or anxiety disorder compared with rates of 11% before the pandemic, with higher prevalence rates for Hispanic compared to non-Hispanic respondents [25,82]. Prevalence estimates for perceived social isolation or loneliness, defined as the discrepancy between an individual’s actual and desired social relationships [83] were 16–43% in the global population during April 2020 [75,84,85,86,87], which represent an increase from pre-pandemic rates [27]. There is some evidence that younger adults may be at greater risk for loneliness during the pandemic than older adults [84].

Psychological stress related to anxiety, depression and/or loneliness is thought to increase breast cancer risk through chronic or repeated activation of the stress-response system (sympathoadrenal and hypothalamic-pituitary-adrenal axes), which can result in hormonal alterations, epigenetic changes, and increased oxidative stress, DNA damage, and inflammation [88,89,90,91,92].

Previous research has suggested that shifting to telework is associated with reduced commuting-related stress [93]. People have also turned to mindfulness techniques in an effort to manage stress during the pandemic. Given that mindfulness meditation and other mindful movement practices (e.g., yoga, tai chi) have also been found to improve depressive symptoms, anxiety, and sleep quality, and reduce inflammation [17,18,19], increased use of mindfulness and meditation programs can improve psychological stress and may reduce the risk of breast cancer.

Perceived social isolation or loneliness [94] may also be a risk factor for breast cancer. Several animal studies have demonstrated that mice and rats that were socially isolated had increased mammary tumorigenesis, or tumor development compared to animals that were not isolation [95,96,97,98,99]. Although only one study has linked loneliness to increased risk for breast cancer in humans [100], several studies have found that loneliness is associated with increased inflammation [101,102,103,104]. As previously cited, inflammation is one of the key drivers of breast cancer risk, and therefore reducing inflammation might have an impact on breast cancer rates. Overall, pandemic-related changes that promote positive health behaviors and reductions in psychological stress and perceived social isolation sequelae might be leveraged for the prevention of breast cancer.

## 4. Leveraging Pandemic Impact for Future Breast Cancer Prevention Research

As described in the preceding sections, there are many impacts of the pandemic on the environment and on individuals that hold the promise of reducing breast cancer risk. Prior to making new policies, clinical guidelines or personal health recommendations, we will need evidence on the feasibility of sustaining of these changes, links between these policies and behaviors and intermediate markers of breast cancer risk, whether short-term increases in stress, sleep disruptions and poor mental health are maintained or change over time to cause lasting effects on cancer risk, and the costs per projected impact on breast cancer rates of these different approaches. 

Since the majority of studies on health behaviors were conducted in the early months of the pandemic and have been cross-sectional in nature, longitudinal studies are needed to assess whether the observed trends in environmental and individual behavior changes are maintained over time. For instance, it is unclear whether individuals will maintain positive health behaviors that they initiated during the pandemic, such as efforts to quit smoking or reduce stress through mindfulness practices.

One of the important findings during the COVID-19 pandemic has been the existence and expression of racial and ethnic disparities in COVID-19 diagnosis, morbidity, and mortality [105,106,107]. Very little of the research described in this chapter has addressed differences in racial and ethnic groups in some of these risk factors. This could be an area of important and relevant research, as we continue to explore the negative and positive effects of COVID-19. 

Given that the majority of studies conducted thus far have focused on the negative impacts of the pandemic on the environment and individual behaviors, another important direction for future research is to investigate resilience factors, including social support, coping strategies, mindfulness, and other protective psychological processes, that may help to promote health and prevent breast cancer. Attention to the pervasive effects of structural racism will be critical to address in new studies and policies.

As the health care environment evolves post-pandemic, it will be necessary to study barriers to implementation and maintenance, and to compare telehealth to traditional office visits for patient-reported outcomes and intermediate cancer risk markers. If effects are similar, that would provide evidence to increase and maintain telehealth, especially since telehealth can reach more people, and potentially could have a greater impact of cancer-related health behaviors, with lower costs.

## 5. Conclusions

Our understanding of the long-term effects of COVID-19 on economic, social, and political events is far from complete. Certainly, this pandemic has hurt several aspects of life in the US and has highlighted existing flaws and disparities in our system that should be addressed. However, our analysis has uncovered several areas that might be seen as improvements in business as usual, and we could consider methods of maintaining or even expanding the positive changes to improve breast cancer prevention efforts. These include the use of telemedicine and improvements in behaviors known to prevent breast cancer. Research is needed to fully understand and use these changes to increase positive behaviors, as well as decreasing negative ones.

## Figures and Tables

**Figure 1 ijerph-18-06999-f001:**
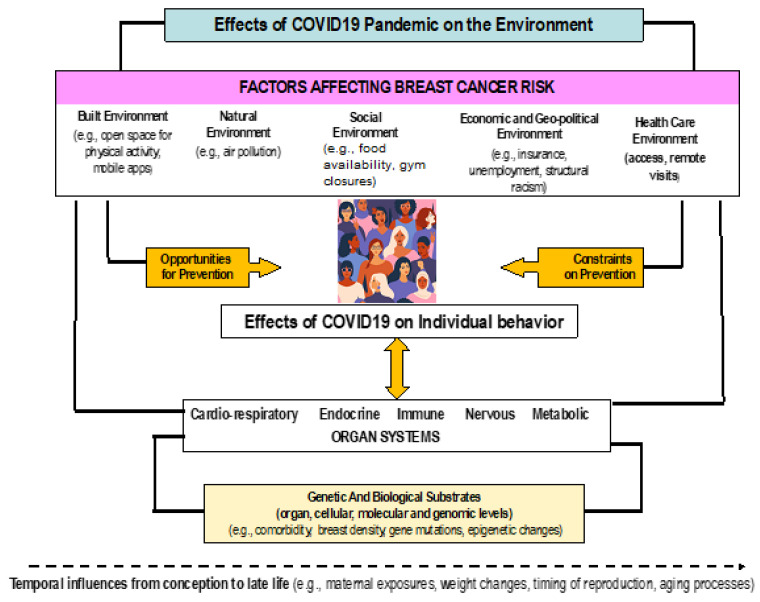
Multi-level Biopsychosocial Model of Influences of the COVID-19 Pandemic on Breast Cancer Prevention and Risk. Adapted from McAtee and Glass [9].

## Data Availability

Not applicable.

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
