# Peer review of "Learning from and Leveraging Multi-Level Changes in Responses to the COVID 19 Pandemic to Facilitate Breast Cancer Prevention Efforts"

_ijerph, 2021, doi:10.3390/ijerph18136999_

Round 1
Reviewer 1 Report
Bowen and coworkers examined the available literature to assess the relationships between COVID-19-related changes and breast cancer risk. The selected topic is clearly current and may be of interest to the readership of the journal. The study takes into account most of the relevant factors that may have an impact on breast cancer incidence, including environmental and individual-level changes and puts forward propositions on how to leverage pandemic impact for breast cancer prevention. The manuscript is well written and takes all relevant categories of epidemiology into consideration. Moreover, consistently bringing resilience factors into the equation is a definite strength of the study. However, the article has several shortcomings, and generally lacks granularity in various aspects of the raised topics.
Points of contention:
- Many of the societal changes may be valid factors contributing to, or protecting the individual from cancer formation, yet these are discussed for the most part as if they were constant overtime. For example, both the shared experience of social isolation, or feelings of social cohesion will undoubtedly develop overtime and will intensify or wane, depending on the extent of the precipitating circumstance. The time factor should be put in perspective.
- In terms of the societal impact of the Covid-19 pandemic an important factor left out of the analysis is age. With retirees, loneliness will develop with a different dynamic and to a different extent than young working adults, or more so in teenagers. Although with breast cancer risk being highest at older age and the overall impact not that severe in the young, long-term repercussions of year-long lockdowns might be worth studying. Age as a factor should be discussed as a modifier to the listed changes in lifestyle.
- Several effects identified in the analysis may strengthen or weaken carcinogenic cues and realizing the dichotomy in these matters is valid. However, for making the assessment of significance the specific, measurable parameters/markers become important, but remain elusive throughout many of the examples listed. Secondly, the size of the effect is also essential for understanding which – the harmful or the beneficial change - has the greater impact.
- The probably most relevant factor in terms of breast cancer development in conjunction with Covid-19 at the level of the individual may be the biologic one. Unfortunately, this aspect has been ignored, while numerous references were made, albeit correctly, to inflammation as an underlying predisposing factor for breast carcinogenesis. Inflammatory cytokines with a central role in the development of Covid-19 –induced pneumonia and post-covid conditions may remain at elevated levels for extended periods of time. Several of those, notably IL-6, IL-8 are well-known factors in mammary tumor formation. Considering the large fraction of symptomatic cases affected along with the mild outcomes with a cytokine response, the epidemiologic impact of these immunologic changes should be discussed, as their impact can be expected to be population-wide.
- A minor point: referenced racial disparities in access to proper care and particularly structural racism might be country-specific or pertain to the US mainly. Given the international reach of the journal this might need to be indicated in the text.
Author Response
Please see file attached for response to reviewers

Reviewer 2 Report
With the current effects of COVID-19 lingering, the authors aim to highlight how changes in the environment and individual behavior might affect the risk of breast cancer. The author’s address these issues in the biopsychosocial framework by breaking up each aspect impacted by COVID-19 in to five categories: built environment, natural environment, social environment, economic and geo-political environment, and health care environment. In each category they address these environmental and individual changes to potential changes in breast cancer risk.
The strength in the article relies in the extensive literature search and how they frame their questions and even provide a graphical representation of their hypothesis. I also feel that the application of the biopsychosocial method in this context is an interesting approach. In most cases this method is used to assess the Health-related quality of life, and how the principles can increase treatment success in patients. With the widespread influence of the epidemic on such a large scale this type of analysis could be beneficial.
One weak point is in their correlation to cancer incidence. I feel that in some of the sections like social environment and economic and geo-political environment the concluding statements on cancer risk could be expanded upon with some reference.
Overall I think this is an interesting analysis and makes some good points that could be investigated in the future.
Author Response
Please see file attached for reply
